# An FDM-Based Dynamic Zoning Method for Disturbed Rock Masses above a Longwall Mining Panel

**Kunyang Zhao [1,2] and Suizi Jia [1,***

[1]   School of Engineering and Technology, China University of Geosciences (Beijing), Beijing 100083, China; zhaoky@cacc.com.cn

[2]   Beijing Super-Creative Technology Co., Ltd., Beijing 100621, China

[*]   Correspondence: 2016010023@cugb.edu.cn

**Abstract:** Underground longwall mining can seriously disturb the surrounding rock masses above the panel. A surface cracking zone, continuous deformation zone, fractured zone, and caved zone can be formed in the overlying strata (termed the "four zones"), which may further result in the spontaneous combustion of coal seams and water inrush. It is essential to study and predict the development characteristics of the four zones induced by longwall mining to guarantee mining safety. These four zones are developed during the mining process, and the mechanical properties in different regions correspondingly differ. Thus, the dynamic zoning characteristics of the disturbed rock masses should be considered in any simulations. In this paper, an FDM-based dynamic zoning method for disturbed rock masses above a longwall mining panel is proposed. This method is mainly composed of four stages: (1) establishing a simplified complete stress-strain curve; (2) determining the zoning criteria; (3) adaptively adjusting the mechanical parameters of the disturbed rock mass; and (4) numerically modeling the longwall mining based on the FDM. The proposed method was applied to a study site in the Taixi coal mine. The dynamic development process of the four zones induced by longwall mining was clearly observed in the modeling procedure. The numerical modeling results achieved in this work, including the periodical coal-seam roof caving and the dynamic development characteristics of the four zones, are consistent with the observed distribution and other studies. The heights of the caved and fractured zones are basically consistent with the empirical formula. Thus, the dynamic zoning method can analyze and predict the dynamic development characteristics of four regions.

**Keywords:** longwall mining; dynamic zoning method; overlying disturbed rock masses; finite difference method

## 1. Introduction

Longwall mining can seriously disturb surrounding rock masses. After mining out the coal seam, deformation, cracking, movement, and caving of overlying strata can occur with the formation of a continuous deformation zone, surface cracking zone, fractured zone, and caved zone (termed the "four zones" [1,2]; see Figure 1). In general, there are many fissures developed in the rock masses of the fractured zone and the caved zone. If the fractured zone develops completely through the upper aquifer, the water in the aquifer can inflow into the mined-out area, which could lead to water inrush. Moreover, if the fractured zone is connected with the surface cracking zone, air leakage channels could be formed and destroy the underground ventilation circulation, which may further lead to the spontaneous combustion of coal seams [3–7]. Therefore, it is essential to study and predict the development characteristics of the "four zones" induced by longwall mining to guarantee mining safety.

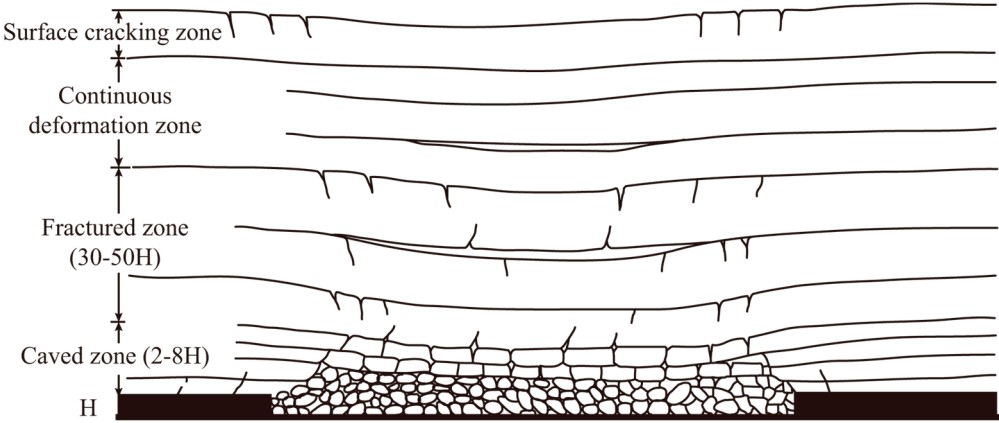

**Figure 1.** The four zones of disturbance in the overlying strata resulting from longwall mining.

Several methods have been employed to study the development characteristics of the four zones in overlying strata, such as field measurements [5,8,9], physical modeling [4,10,11], and numerical modeling [3,6,7,12–14]. Numerical modeling can be considered a promising and effective tool to reflect the overburden movement process and predicting the heights of disturbed zones in longwall mining. Various examples of numerical modeling have been reported in the literature. Researchers have used different numerical modeling methods, including the finite element method (FEM) [12], finite difference method (FDM) [7,10], discrete element method (DEM) [3,6], and mesh free method (MFM) [13].

There are several types of zoning methods introduced in the literature. The commonly used zoning methods are outlined below.

In the first method, rock masses are considered elastoplastic materials while employing a strain-softening model, and the process of caving is simulated after determining the parameters in the strain-softening model when rock masses fail using an empirical formula [12,15]. As the failed rock masses freely fall from the roof during caving, Ghabraie et al. assumed that the stresses of the rock masses suddenly drop to almost zero after failure [12]. After satisfying the yielding condition, the material fails, and its properties are updated to those of the damaged material. In this method, the differences in the mechanical characteristics of roof rock masses before and after caving are considered, and it is feasible to simulate strata caving by adjusting the mechanical characteristics of failed rocks. However, this method considers rock mass yielding as the caving criterion without clearly defining the scope of the caved zone, which may lead to a misjudgment of the scope.

In the second method, the heights and shapes of the caved zone and fractured zone are calculated according to an empirical formula, and then the scopes of the two zones above the mined-out area and the corresponding mechanical parameters are determined before conducting a simulation of longwall mining. Tajduś [16] assumed that the caved zone existed with a height *hz* and a shape similar to a trapezoid above the mined-out area. Moreover, the fractured zone existed with a height *hs* and a shape similar to a half-ellipse above the caved zone. The heights depended on the mechanical properties of the rock mass layers and the thickness of the excavation. The parameters of the two zones were determined with laboratory tests and the GSI (geological strength index). In this method, the differences between different zones are considered, and the scope of disturbed rock masses is clearly determined. However, the determination of the scope of the caved zone and the fractured zone is strongly impacted by expert judgments, and the roof strata development process from the elastic-plastic deformation stage to the cracking and falling stage is ignored.

In the third method, the height of the caved zone is determined according to the empirical bulking factor, and in the simulation, if the strain or displacement of the falling rock mass reaches the caving criterion, it is considered a new loose material. Shabanimashcool and Li used the principal plastic strain to identify the caving boundary and the bulking

factor to determine whether the roof strata of the panel entirely filled the mined-out area. After the caving process stops, the mechanical properties of caving materials are changed to a double-yield model [17,18]. In this method, the caving criterion is used to determine whether the surrounding rock falls in a mining simulation, which avoids misjudging the yielded noncaving rock as caved rock. The roof strata development process from elastic-plastic deformation to cracking and caving can be dynamically reflected. Moreover, the change in mechanical parameters of caved rock can also be well reflected. However, in this method, the empirical formula is used to pre-set the maximum height of the caved zone, which may be strongly affected by human intervention.

In the fourth method, caving criteria are established according to experience. In the simulation of the mining process, the boundary of the caved zone is determined dynamically according to the caving criterion, and the mechanical parameters of the rock masses in the caved zone are adjusted according to experience. Singh and Singh thought that failure of the strata was the preliminary requirement for its caving [19–21]; therefore, the criterion for caving of the failed strata (failure in tension) to occur in the model was a maximum shear strain of 0.25 or a downwards displacement of 1 m. Then, the cohesion value of caved materials was set to zero. This method does not need to pre-set the height of the caved zone; thus, it can avoid the calculation error caused by human intervention. However, the zoning criteria also have limitations. In reality, the critical shear strain is strongly site-dependent, and the vertical displacement of the caved rock masses is associated with the size of the mined-out area.

In most of the currently used methods for zoning simulation, specific attention is usually paid to modeling the process of roof rock falling and the adjustment of caved rock mass parameters, while in contrast, the strength deterioration of rock masses in other zones is neglected. Although the rock masses in the fractured zone, continuous deformation zone, and surface cracking zone are not broken or collapsed, the strength deterioration of rock masses varies with the extent of damage. In addition, the four zones are developed during the mining process, and the mechanical properties in different regions correspondingly differ. The dynamic zoning characteristics of disturbed rock masses should not be neglected; otherwise, inexact numerical results may be achieved.

In this paper, an FDM-based dynamic zoning method for disturbed rock masses above a longwall mining panel is proposed. This method is mainly composed of four stages: (1) establishing the simplified complete stress-strain curve; (2) determining the zoning criteria; (3) adaptively adjusting the mechanical parameters of disturbed rock masses; and (4) numerically modeling longwall mining based on the FDM. To demonstrate the effectiveness of the proposed dynamic zoning method, a numerical application case is presented with the use of FLAC$^{3D}$ software. FLAC$^{3D}$ is inherently capable of solving nonlinear and large-strain problems; in particular, it can be well applied to modeling the distribution patterns of the stress, strain, and displacement of surrounding rock masses in underground mining [22].

## 2. The Dynamic Zoning Method

### 2.1. Overview of Our Method

During the underground mining of coal seams, the overlying disturbed rock mass undergoes all or part of the four stages of elastic-plastic deformation, cracking, caving, and caved material deformation. Correspondingly, we simplify the complete stress-strain curve of the disturbed rock mass into four stages: the elastic deformation stage (Stage I), post-peak softening stage (Stage II), residual strength stage (Stage III), and caved material deformation stage (Stage IV).

According to the deformation and failure characteristics of the rock mass in different stages, the zoning criteria of the rock mass are established: the rock mass in the elastic deformation stage and post-peak softening stage belongs to the continuous deformation zone; the rock mass in the residual strength stage belongs to the fractured zone or the

surface cracking zone; and the rock mass in the caved material deformation stage belongs to the caved zone.

According to the simplified complete stress-strain curve, the mechanical parameters of the disturbed rock mass are adaptively adjusted: the parameters are maintained in Stage I, reduced in Stage II, adjusted to satisfy the residual state in Stage III, and adjusted to satisfy the condition of loose deposits in Stage IV.

According to the damage evolution process of rock mass deformation in actual underground mining, by considering the mining process simulation, rock mass zoning criteria, and adaptive adjustment of rock mass mechanical parameters, the simulation of a longwall mining process is conducted based on the FDM.

By exploiting the abovementioned four stages, the dynamic zoning of the disturbed rock masses can be realized (Figure 2).

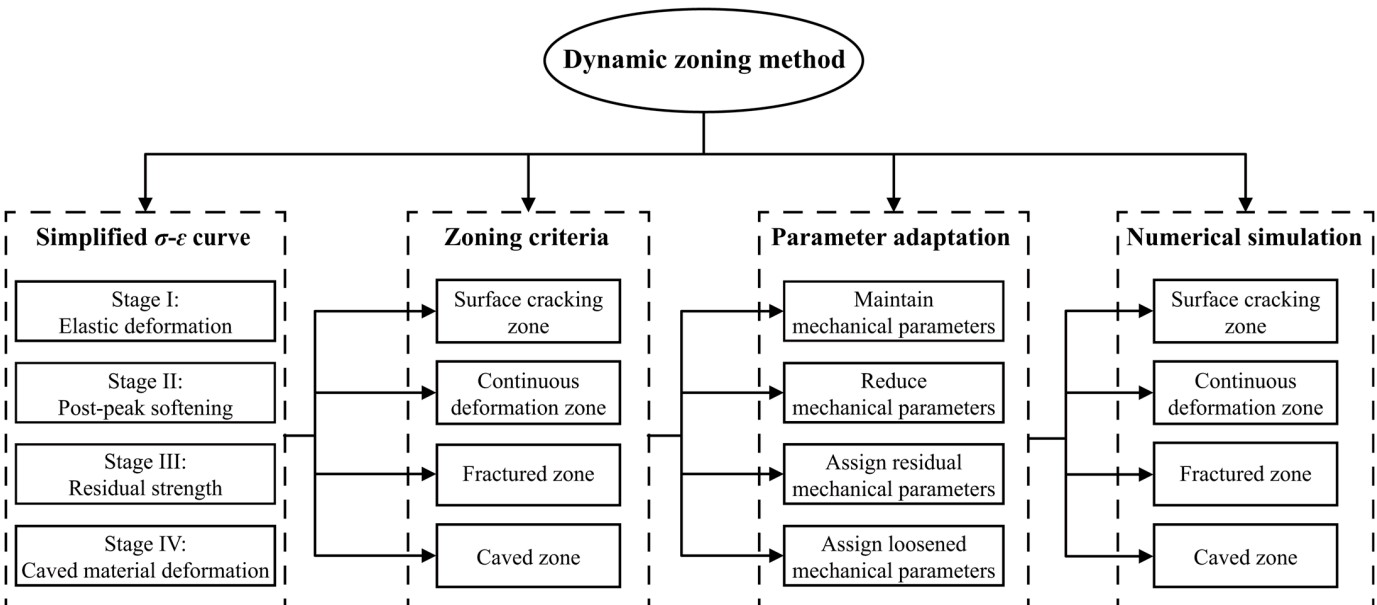

**Figure 2.** Outline of the dynamic zoning method.

*2.2. Establishing the Simplified Complete Stress-Strain Curve*

In general, there are several common characteristics for rock mass complete stress-strain curves [23–25] (Figure 3). (1) Before reaching the peak strength, the rock mass is in the elastic deformation stage, and the deformation at this stage gradually develops with increasing stress. (2) After reaching the peak strength, the rock mass is in the softening stage. With increasing deformation, the strength of the rock mass gradually decreases. (3) After reaching the residual strength, the rock mass is in the residual strength stage. With the continuous development of plastic deformation, the strength of the rock mass remains at a residual value. (4) With increasing confining pressure, the peak strength and residual strength of the rock mass gradually increase, and the peak strain and residual strain increase correspondingly.

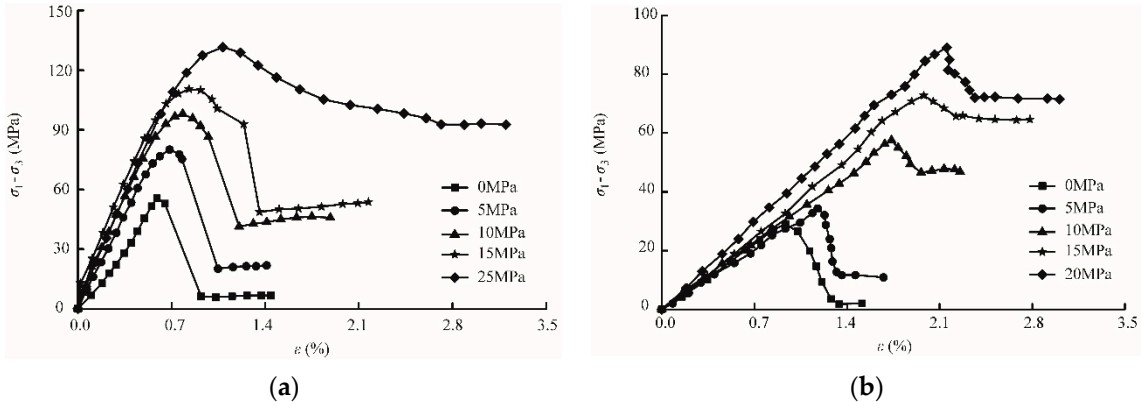

**Figure 3.** Complete stress-strain curves of rock: (**a**) sandstone; (**b**) mudstone.

On the basis of the above characteristics, this paper simplifies the complete stress-strain curve of the overlying disturbed rock mass into four stages (Figure 4b). Stage I (Section OA) is the elastic deformation stage, i.e., $\varepsilon^p = 0$; Stage II (Section AB) is the post-peak softening stage, i.e., $0 < \varepsilon^p < \varepsilon^p_{residual}$; Stage III (Section BC) is the residual strength stage, i.e., $\varepsilon^p_{residual} \leq \varepsilon^p < \varepsilon^p_{loosened}$; and Stage IV (Section CD) is the caved material deformation stage, i.e., $\varepsilon^p \geq \varepsilon^p_{loosened}$. More details are shown in Figure 4.

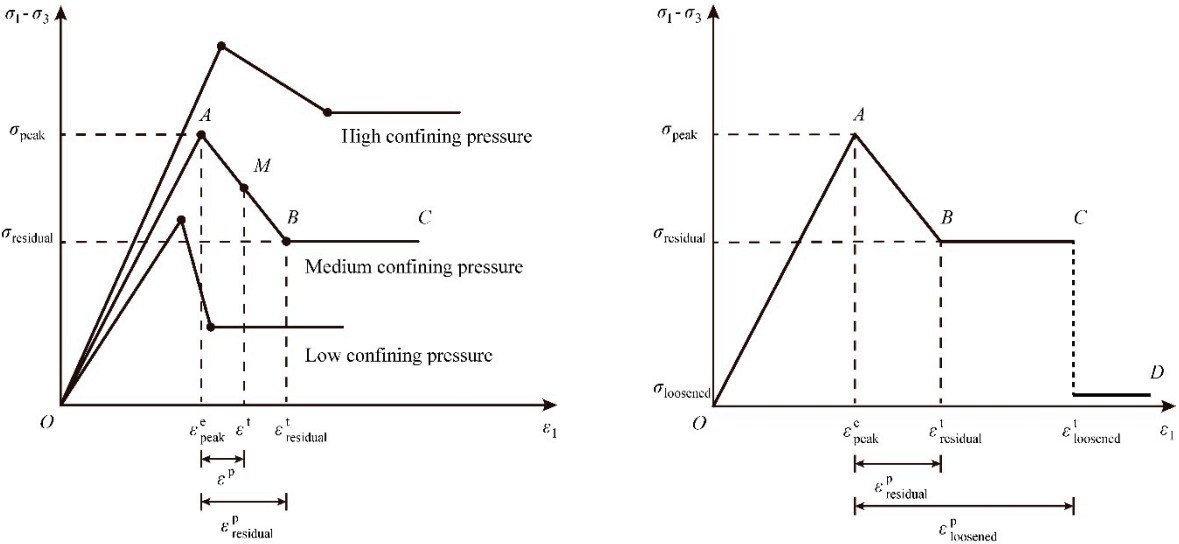

(**a**) Simplified stress-strain curve  (**b**) Stress-strain curve of overlying disturbed rock mass

**Figure 4.** The simplified complete stress-strain curve of the rock mass.

(1) Elastic deformation stage (Stage I): Before the rock mass yields, its strain develops linearly with increasing stress; see Section OA of the simplified complete stress-strain curve (Figure 4a).

(2) Post-peak softening stage (Stage II): After the rock mass yields, internal cracks gradually develop, expand, and coalesce. With increasing rock mass strain, the strength of the rock mass degenerates; see Section AB of the simplified stress-strain curve (Figure 4a). The corresponding strength of Point A is the peak strength $\sigma_{peak}$, and the corresponding strain is the maximum elastic strain $\varepsilon^e_{peak}$; the corresponding strength of Point B is the residual strength $\sigma_{residual}$, and the corresponding strain is $\varepsilon^t_{residual}$. See Equation (1):

$$\varepsilon^t_{residual} = \varepsilon^e_{peak} + \varepsilon^p_{residual} \tag{1}$$

For any point M located on Section AB (Figure 4a), its strain should satisfy Equation (2):

$$\varepsilon^t = \varepsilon^e_{peak} + \varepsilon^p \tag{2}$$

In addition, when subjected to different confining pressures, both the complete stress-strain curves and their simplifications are different. With increasing confining pressure, the peak strength $\sigma_{peak}$ and residual strength $\sigma_{residual}$ of the rock mass gradually increase, and the corresponding peak strain $\varepsilon^e_{peak}$ and residual strain $\varepsilon^t_{residual}$ also increase gradually (Figure 4a).

$\varepsilon^p_{residual}$ represents the threshold plastic strain of the rock mass from the post-peak softening stage to the residual strength stage, which depends on the property and stress state of the rock mass, and it is difficult to obtain the value of $\varepsilon^p_{residual}$ directly. As $\varepsilon^e_{peak}$ and $\varepsilon^t_{residual}$ change with increasing confining pressure (Figure 4a), the critical value of the plastic principal strain $\varepsilon^p_{residual}$ also changes with the confining pressure.

Assuming that the relationship between $\varepsilon^p_{residual}$ and the confining pressure is piecewise linear (Figure 5), the critical value of the plastic principal strain $\varepsilon^p_{residual}$ subjected to a certain confining pressure can be determined according to the simplified complete stress-strain curve. The continuous relation curve of $\varepsilon^p_{residual} - \sigma_3$ can be gained by simply connecting two adjacent points with a straight line, and thus the critical values of the plastic principal strain under different confining pressures can be calculated.

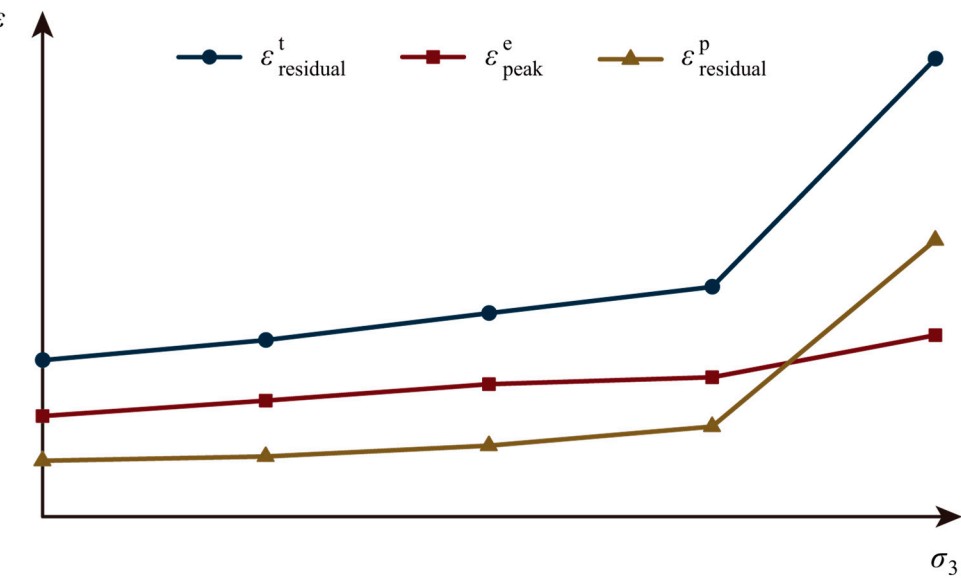

**Figure 5.** Diagram of the strain-confining pressure relationship.

(3) Residual strength stage (Stage III): The rock mass cracks and reaches the residual strength, but it can remain in the layered form. With the continuous development of deformation, the rock mass strength remains at the residual value; see Section BC of the simplified complete stress-strain curve (Figure 4a).

(4) Caved material deformation stage (Stage IV): When the rock mass completely fails and breaks into caved material, i.e., the rock mass is in the stage of caved material deformation, the strength of the caved material is slower than the residual strength of the rock mass and assumed to be unchanged; see Section CD of the simplified stress-strain curve (Figure 4a). The corresponding strength of Point C is the strength of the caved material $\sigma_{loosened}$, and the corresponding strain is $\varepsilon^t_{loosened}$, which satisfies Equation (3):

$$\varepsilon^t_{loosened} = \varepsilon^e_{peak} + \varepsilon^p_{loosened} \tag{3}$$

$\varepsilon_{loosened}^{p}$ represents the critical plastic strain of the rock mass from being in the residual strength stage to the caved material deformation stage. The strains at which strength loss occurs are derived based on an investigation of parameters used for the modeling of similar problems in the literature [17–20,26]. It can be summarized that the cohesion and friction properties of rock mass are assumed to drop to a minimum residual value at a plastic shear strain larger than 0.05 (5% strain), and the tensile strength falls to a residual value of zero at a plastic tensile strain larger than 0.01 (1% strain).

*2.3. Determining the Zoning Criteria*

In the process of underground mining, the deformation and failure characteristics of the overlying disturbed rock masses are different, which corresponds to the different stages of the simplified complete stress-strain curves. The rock masses in the continuous deformation zone undergo elastic-plastic deformation, and they are in the elastic deformation stage (Stage I) or post-peak softening stage (Stage II). The rock masses in the surface cracking zone and the fractured zone go through the process including elastic-plastic deformation and cracking, and they are in the residual strength stage (Stage III). After the process of elastic-plastic deformation, cracking, and caving, rock masses in the caved zone fill the mined-out area; the caved rocks are in the caved material deformation stage (Stage IV).

According to the simplified complete stress-strain curve, the zoning criteria are established by considering the deformation and failure characteristics, as well as the location of the overlying disturbed rock masses.

The parameter $\varepsilon^p$, which represents the plastic strain of the rock mass, is one of the major metrics for dynamic zoning, while ' denotes the total strain of the disturbed rock mass. Details of the zoning criteria are listed as follows.

(1) $\varepsilon^p = 0$ and $\varepsilon^t \geq 0$.

This means that the rock mass is in the elastic deformation stage, and if the rock mass is located above the fractured zone, it will be classified into the continuous deformation zone.

(2) $0 < \varepsilon^p < \varepsilon_{residual}^{p}$.

This means that the rock mass is in the post-peak softening stage, and if the rock mass is located above the fractured zone, it will be classified into the continuous deformation zone.

(3) $\varepsilon_{residual}^{p} \leq \varepsilon^p < \varepsilon_{loosened}^{p}$.

This means that the rock mass is in the residual strength stage. If the rock mass is located above the caved zone, it will be classified into the fractured zone, whereas if the rock mass is located near the ground surface, it will be classified into the surface cracking zone.

(4) $\varepsilon^p \geq \varepsilon_{loosened}^{p}$.

This means that the rock mass is in the caved material deformation stage, and if the rock mass is located above the mine mined-out area, it will be classified into the caved zone.

It should be noted that in general, a rock mass is of low tensile strength, and if the rock mass is subjected to tensile failure, it should be considered to be in the residual strength stage.

*2.4. Adaptive Adjustment of Mechanical Parameters of Disturbed Rock Masses*

In the process of underground mining, the deformation and failure characteristics of the overlying rock masses dynamically vary, and the mechanical properties correspondingly change. With the mining of the coal seam, at first, the overlying disturbed rock mass undergoes small deformation, and the rock mass is in the elastic deformation stage. After the rock mass yields, internal fissures occur, and the rock mass gradually loses continuity, but it can still maintain the layered form. At this time, the rock mass is in the post-peak softening stage, and the strength parameters (cohesion $c$ and friction angle $\varphi$) are gradually reduced. The cohesion $c$ and friction angle $\varphi$ tend to stabilize until the rock masses reach the residual strength stage. When the coal-seam roof begins to cave and fill into the mined-out area, the rock masses are in the caved material deformation stage, and the cohesion $c$ and friction angle $\varphi$ reduce to the parameters of the caved materials.

According to the simplified complete stress-strain curve, the adaptive adjustment of the mechanical parameters of rock masses at different stages is as follows:

(1) Elastic deformation stage (Stage I): the mechanical parameters of rock masses in this stage remain unchanged.

(2) Post-peak softening stage (Stage II): the mechanical parameters of rock masses in this stage need to be reduced.

The principles of damage mechanics are employed to interpret the characteristics of the post-peak softening stage of the disturbed overlying strata. More specifically, the formation, expansion, and coalescence of cracks in rock masses can be considered a damage process of the mechanical properties of rock masses. That is, the post-peak softening stage of the coal-seam roof can be regarded as a progressive damage process along with the development of plastic strains.

In the post-peak softening stage, with increasing plastic strain $\varepsilon^p$, the cohesion $c$ decreases from an initial value $c_{initial}$ to a residual value $c_{residual}$, and the friction angle $\varphi$ decreases from an initial value $\varphi_{initial}$ to a residual value $\varphi_{residual}$.

More specifically, $\varepsilon^p_{residual}$ can be determined according to the confining pressure $\sigma_3$ (Figure 5); thus, the linear softening curves of cohesion $c$ and friction angle $\varphi$ are determined (Figure 6). According to the $\varepsilon^p$ of any point in the post-peak softening stage, the corresponding cohesion $c_{new}$ and friction angle $\varphi_{new}$ can be achieved.

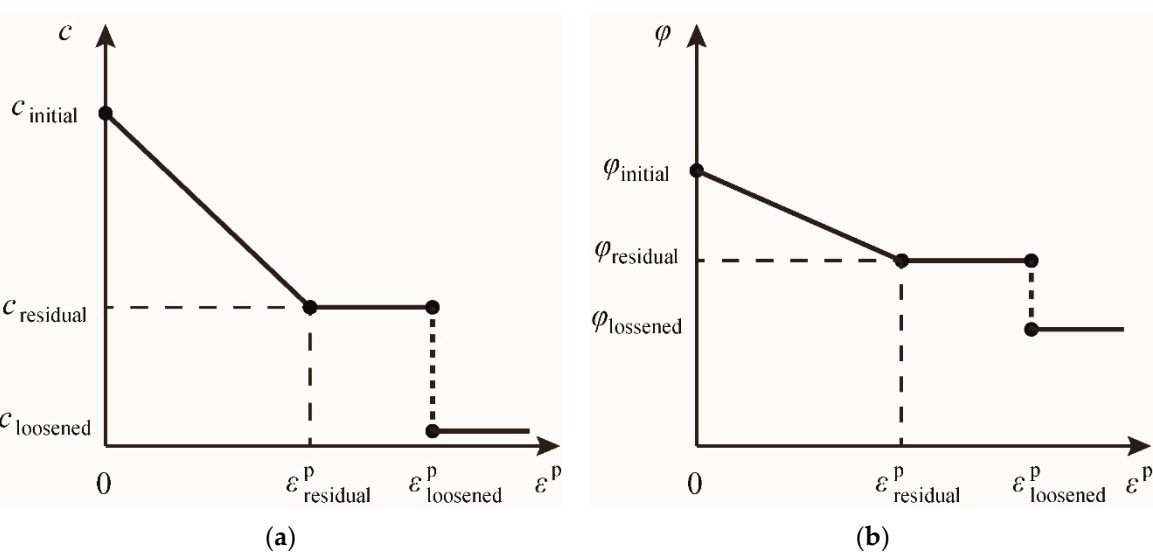

**Figure 6.** Softening model of strength parameters: (**a**) softening model of cohesion; (**b**) softening model of friction angle.

(3) Residual strength stage (Stage III): the mechanical parameters of rock masses in this stage are adjusted to those parameters in the residual state (i.e., $c_{residual}$ and $\varphi_{residual}$).

In the residual strength stage, with increasing plastic strain, the cohesion $c$ and friction angle $\varphi$ remain at residual values of $c_{residual}$ and $\varphi_{residual}$, respectively (Figure 6).

(4) Caved material deformation stage (Stage IV): the mechanical parameters of rock masses in this stage are adjusted to those parameters of the caved materials (i.e., $c_{loosened}$ and $\varphi_{loosened}$).

When the roof rock masses fall and break into irregular blocks, the cohesion of the rock mass decreases significantly, while the friction angle changes little. In some studies [17–20], the cohesion of the caved material is reduced to zero, and the friction angle is not changed or is maintained at the residual value. In this paper, it is assumed that $c_{loosened} = 0$ and $\varphi_{loosened} = \varphi_{residual}$ (Figure 6).

### 2.5. Numerical Modeling of Longwall Mining Based on the FDM

The four zones are developed during the actual mining process, and the mechanical properties in different zones correspondingly change. Therefore, the proposed dynamic zoning method considers the mechanical parameter adaptation of disturbed rock masses in the progressive mining process. More specifically, according to the simplified complete stress-strain curve, the zoning criteria are established by considering the deformation and failure characteristics of the disturbed rock masses, and the mechanical parameters of the disturbed rock masses are adaptively adjusted. On those bases, the dynamic zoning of the disturbed rock mass in the mining process is simulated, and the dynamic development characteristics of the four zones are obtained.

This paper simulates the underground longwall mining process with FLAC$^{3D}$ by means of step-by-step excavation. More details of the simulation process are introduced as follows (Figure 7):

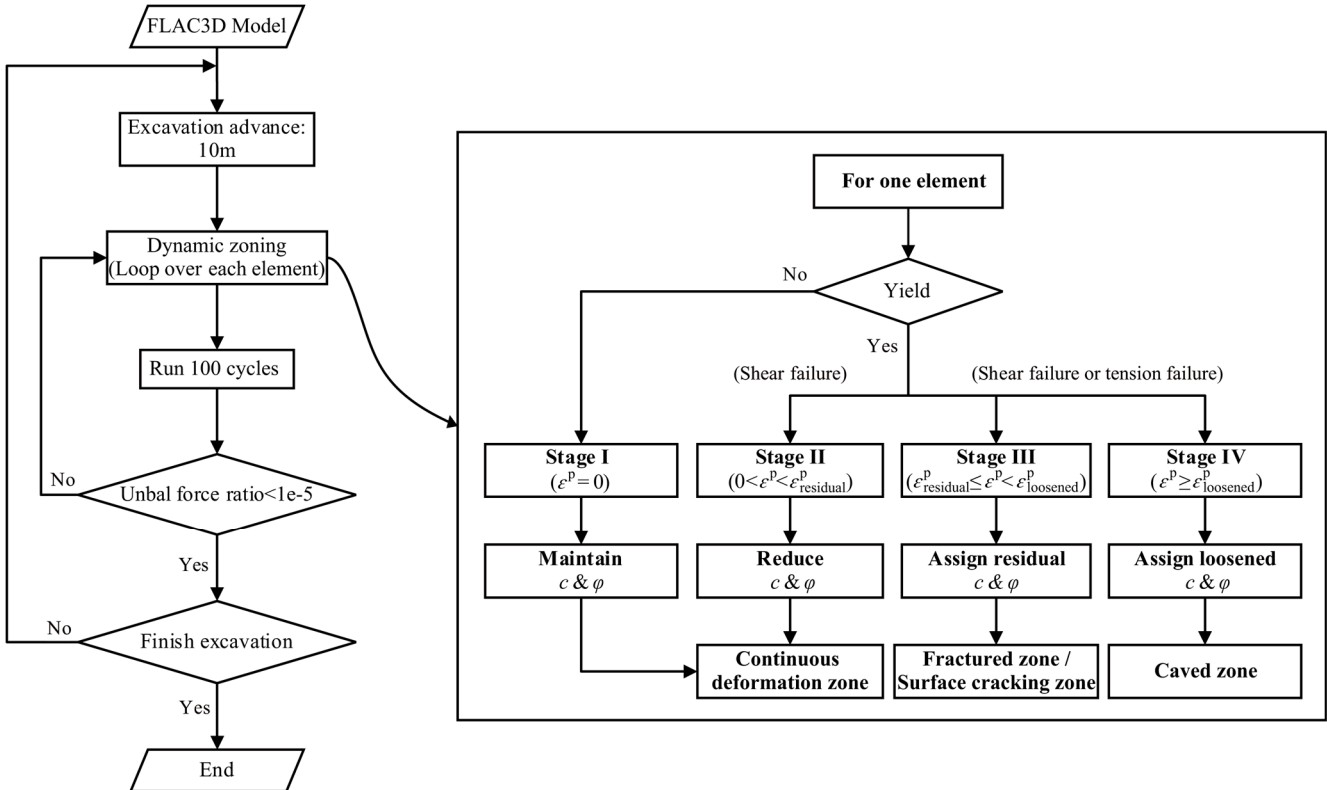

**Figure 7.** Flowchart for the simulation of longwall mining and dynamic zoning processes.

Step 1. Mining with a distance of L in each step.

L is determined according to the actual mining schedule; in this paper, L = 10 m.

Step 2. Adaptively adjusting the parameters of the overlying disturbed rock masses and dynamically zoning by looping over all the elements of the computation model.

For each element, it is first necessary to determine whether the element yields. If not, then the element is in the elastic deformation stage; otherwise, the element needs to be further checked according to the failure type and plastic strain of the element to determine which stage it is in (the post-peak softening stage, the residual strength stage, or the caved material deformation stage). Second, by exploiting the principles of adaptively adjusting the mechanical parameters of rock masses in different stages, the cohesion $c$ and the friction angle $\varphi$ of the element are adjusted correspondingly. Finally, by following the zoning criteria, it must be determined which zone the element belongs to.

Step 3. Checking whether the numerical computation reaches computational balance after N iterations (in this paper, N = 100).

If the computation is balanced (i.e., the maximum unbalanced force ratio is less than $1 \times 10^{-5}$), then it should further conduct mining with distance L and repeat Step 2; otherwise, it needs to repeat Step 2 directly until the computation is balanced.

Step 4. Proceeding to the next step of mining and repeating Steps 1–3.

The mechanical parameters of the disturbed rock masses are dynamically adjusted, and the scope of the four zones is updated in real time until the end of the mining.

The program code of the algorithm (Figure 7) is written by using the FISH language in FLAC3D software, and it is embedded in the numerical simulation calculation process to realize the dynamic adjustment of the mechanical parameters of the disturbed strata. See Appendix A for the algorithm code.

Additionally, the following methods are used to simulate coal-seam mining: first, before FLAC3D simulation mining, the interfaces are set at the contact surface between the coal seam and the roof, which can avoid the dragging effect on the contact surface when the coal seam is mined, and prevent the distortion of the grid displacement data near the interface, so as to realize the full collapse of the coal-seam roof and the natural collapse to the floor and ensure the numerical simulation effect. Then, according to the coal-seam mining step, the coal-seam units within the step are emptied (that is, set to null model) to simulate the coal-seam excavation in the current step. Finally, it is iterated to the equilibrium state (the maximum unbalance ratio is less than $1 \times 10^{-5}$), the calculation model is saved, and the next mining step according to the above method is then simulated until the coal-seam excavation of the working face is completed.

## 3. Case Study

### 3.1. Background of the Study Site

3.1.1. Geological Setting

The Taixi coal mine is located in Jining City, Shandong Province, China. The territory of the area is flat, and the ground elevation is 38~41 m, with a slight fluctuation difference of less than 3 m (see Figure 8). It has been revealed through drilling that the strata from the newest to oldest in this area are Quaternary (Pingyuan formation), Neogene (Minghuazhen formation), Permian (Shihezi and Shanxi formations), and Carboniferous (Taiyuan formation); see Figure 9. More details are as follows.

(1) The Quaternary ($Q_p$) stratum is mainly composed of yellow clay loam intercalated with multilayer sand. The stratum has an average thickness of approximately 215.55 m and is in unconformable contact with the Neogene.

(2) The Neogene ($N_{2m}$) stratum is mainly composed of red, yellow-brown mudstone and fine–coarse sandstone. The stratum has an average thickness of approximately 152.85 m and is in unconformable contact with the Permian.

(3) The Permian (P) is divided into the following two formations, from top to bottom.

The Shangshihezi formation ($P_{1+2s}$) is mainly composed of grey, purple, and other variegated mudstones and sandstones, and the bottom is carbonaceous mudstone. The stratum is in continuous deposition with the underlying Shanxi formation, with an average thickness of approximately 167.8 m.

The Shanxi formation ($P_{1x}$) is composed of black mudstone and greyish white sandstone. The average thickness of the Shanxi formation is approximately 52.94 m. The #3 coal seam is located in the lower part of the Shanxi formation, with an average thickness of approximately 3.5 m.

(4) The Carboniferous ($C_{2t}$) stratum is mainly composed of green-grey siltstone, mudstone, grey-white medium sandstone, fine sandstone, and limestone. The average thickness of this formation is approximately 174.69 m.

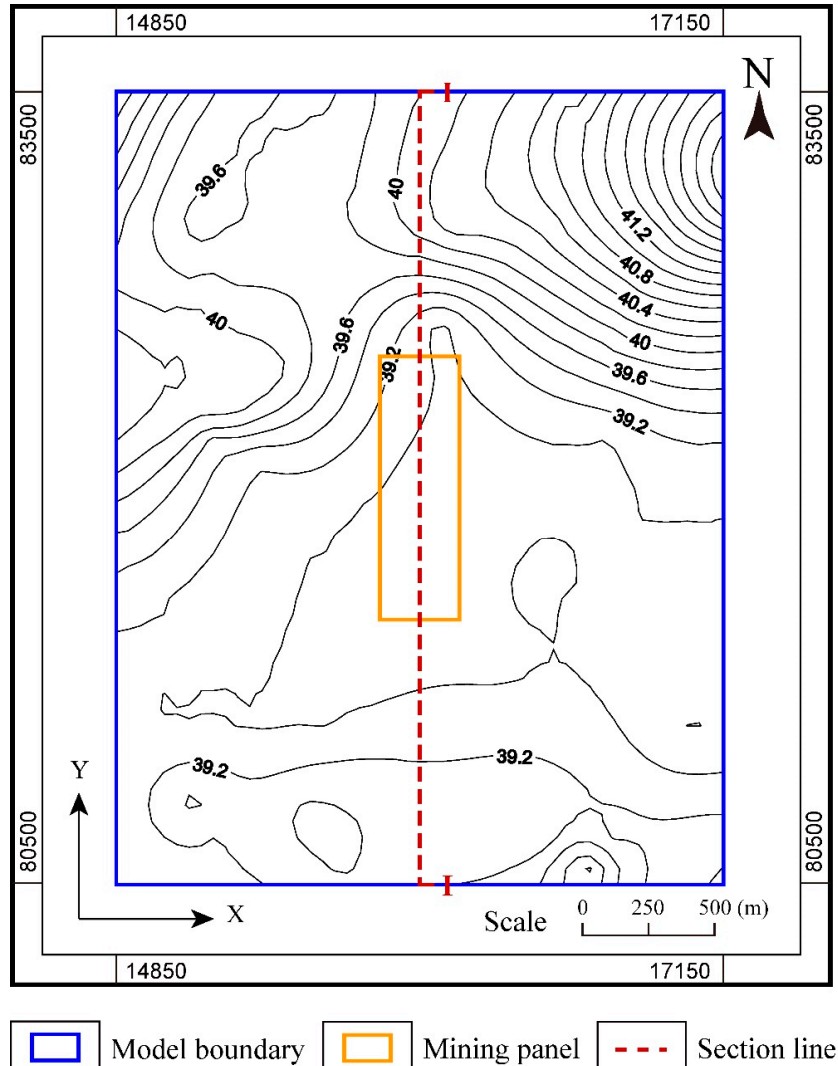

**Figure 8.** The contour map of the ground surface in the study area.

| Stratigraphic unit | | Thickness | Remarks |
|---|---|---|---|
| System | Formation | | |
| Quaternary | $Q_p$ | 215.6m | Aquifer |
| Neogene | $N_{2m}$ | 152.9m | Aquifuge |
| Permian | $P_{1+2s}$ | 167.8m | |
| | $P_{1x}$ | 26.0m | Coal seam roof |
| | | 3.5m | 3# coal seam |
| | | 23.6m | Coal seam floor |
| Carboniferous | $C_{2t}$ | 175.0m | |

**Figure 9.** Stratigraphic column of the study site.

### 3.1.2. Mine Layout

The panel is 300 m wide and 1000 m long; see Figure 8. The average thickness of Coal Seam 3 in the mining area is 3.5 m, and the mining depth varies from 550 to 570 m. Coal is mined using top coal mining technology. The mining direction is from south to north.

### *3.2. Computational Model*

#### 3.2.1. Model Domain

The domain of the calculation model is determined based on the mining area size of the panel; see Figure 8. For simplicity, we have created a local coordinate system, setting the north–south direction shown in Figure 8 as the *Y* axis, setting the east–west direction as the *X* axis, and setting the elevation direction as the *Z* axis; see Figure 10. The plane boundary of the calculation model is simply obtained by extending the boundary of the panel along the west–east and south–north directions for 1000 m. The distance along the west–east direction is 2300 m, while the distance along the south–north direction is 3000 m. The altitude range is between −720 and 44 m. In the plan view, the coordinates of the southwest corner and the northeast corner are (148508500) and (171508500), respectively.

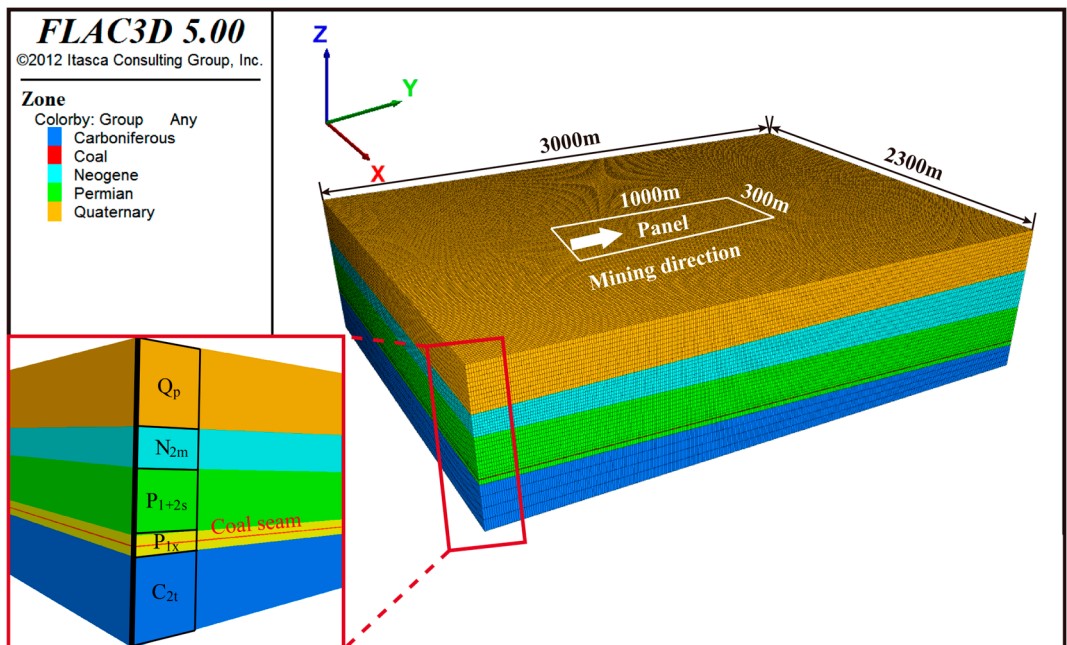

**Figure 10.** Computational mesh model.

#### 3.2.2. Generalization of Strata

The strata of the three-dimensional geological model from top to bottom are: Quaternary, Neogene, Permian, and Carboniferous; see Figure 9. The geometric characteristics of each formation are determined by borehole logs and geological maps.

#### 3.2.3. Geological Model

The geological model is meshed using a set of hexahedral elements. FLAC3D allows a variety of shape options, including hexahedron, tetrahedron, pyramid, triangular prism, and so on (Itasca Consulting Group, Inc., Minneapolis, MN, USA, 2009). Hexahedral elements are usually the best choice for numerical simulation.

The presented computational mesh model is composed of 2,691,000 elements and 2,781,240 nodes. The model is divided into 39 layers vertically with an overburden layer having a thickness of 4.0~19.6 m. The closer a layer is to the coal seam, the thinner the layer is set. In the horizontal direction, each layer is meshed into 69,000 squares and 69,531 nodes. The squares have an edge length of approximately 10 m; see Figure 10.

### 3.2.4. The Constitutive Model and Yield Criterion

The Mohr Coulomb yield criterion and elastoplastic constitutive model were used in this case study.

### 3.2.5. Boundary Conditions

Displacement boundary conditions were used in this study [22]. The boundary conditions of the model are set as follows: (1) applying fixed constraints in the X direction to the east and west boundaries of the model; (2) applying a fixed constraint in the Y direction to the southern and northern boundaries; and (3) applying fixed constraints in the Z direction to the bottom of the model. In addition, no constraints are added to the upper surface of the model; see Figure 10.

### 3.2.6. Calculation Parameters

The physical and mechanical parameters of the rock mass in the study area were determined through physical and mechanical experiments of the rock mass; evaluation parameters of the rock mass quality are listed in Table 1. Each formation often contains several rock strata with different lithology. For simplicity, we used the weighted average method to assign a set of parameters to each formation. We first determined the cohesion, internal friction angle, modulus of elasticity, Poisson's ratio, gravity, and other parameters of the main rocks in each rock stratum according to the mechanical experiments, and then determined the parameters of each formation according to the thickness of the rock stratum and the above parameter values using the weighted average method.

**Table 1.** The adopted physical–mechanical parameters.

| Strata | Rock Formation | Elastic Modulus (GPa) | Poisson's Ratio | Density (kg/m$^3$) | Initial | | | Residual | | |
|---|---|---|---|---|---|---|---|---|---|---|
| | | | | | Cohesion (MPa) | Friction Angle (°) | Tensile Strength (MPa) | Cohesion (MPa) | Friction Angle (°) | Tensile Strength (MPa) |
| Q | | 0.05 | 0.32 | 1980 | 0.06 | 23 | 0.02 | 0.01 | 23 | 0.002 |
| N | | 3.27 | 0.30 | 2340 | 0.21 | 25 | 0.07 | 0.12 | 19.8 | 0.033 |
| P | P$_{1+2s}$ | 10.20 | 0.23 | 2517 | 0.97 | 34 | 0.37 | 0.09 | 19.3 | 0.025 |
| P | P$_{1x}$ | 10.90 | 0.21 | 2650 | 0.97 | 37 | 0.39 | 0.10 | 18.5 | 0.027 |
| P | Coal seam | 0.35 | 0.34 | 2350 | 0.17 | 20 | 0.05 | - | - | - |
| C | C$_{2t}$ | 14.80 | 0.28 | 2700 | 0.87 | 27 | 0.28 | - | - | - |

In this paper, $\varepsilon^p_{residual}$ of the overlying strata is determined by the simplified complete stress-strain curve of the main lithology. Taking the Shanxi formation (P$_{1x}$) as an example, the determination process of $\varepsilon^p_{residual}$ is briefly described. The Shanxi formation (P$_{1x}$) is mainly composed of sandstone and mudstone. First, the complete stress-strain curves of sandstone and mudstone are obtained by triaxial tests (see Figure 3), which are simplified as ideal curves, as shown in Figure 11, and $\varepsilon^p_{residual}$ of sandstone and mudstone under various confining pressures are obtained. Second, based on the thicknesses of sandstone and mudstone in the formation, the weighted average value $\varepsilon^p_{residual}$ of the Shanxi formation under each confining pressure is calculated. Finally, the $\varepsilon^p_{residual} - \sigma_3$ relation curve (Figure 12) can be drawn, and the $\varepsilon^p_{residual}$ corresponding to arbitrary confining pressure $\sigma_3$ can be obtained on the basis of the curve.

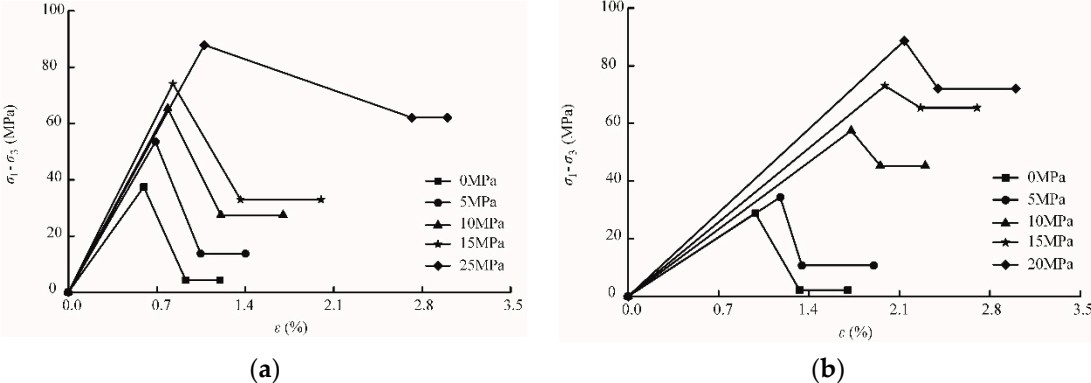

**Figure 11.** Simplified stress-strain curve: (**a**) sandstone; (**b**) mudstone.

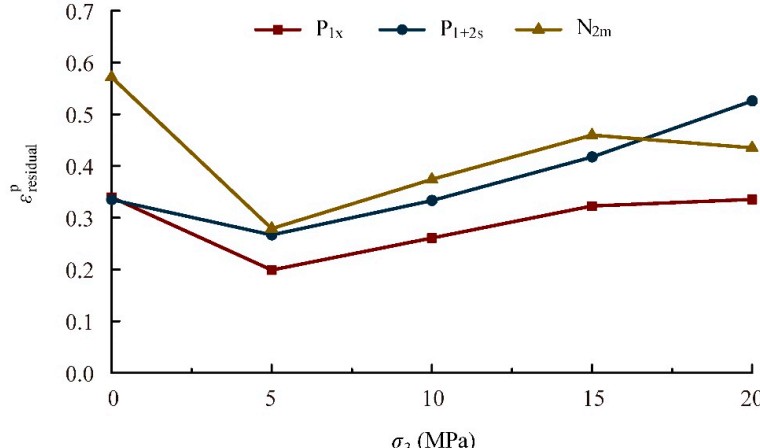

**Figure 12.** Schematic diagram of the relationship between the residual critical plastic strain and confining pressure.

According to the above method, the $\varepsilon^p_{residual} - \sigma_3$ relationship curves of all the overlying strata can be obtained (Figure 12).

At the study site, the complete stress-strain curve of the yellow clay loam in the Quaternary (the Pingyuan formation) shows weak softening. With the continuous development of deformation, the yellow clay loam has little stress variation, and the strength of the yellow clay loam remains at the residual value. Therefore, it is considered that the yellow clay loam would be in the residual strength stage when $\varepsilon^t > \varepsilon^e_{peak} (\varepsilon^p_{residual} = 0)$.

The $\varepsilon^p_{loosened}$ is determined according to the empirical values in the references, $\varepsilon^p_{loosened} = 0.14$ (in tension) or $\varepsilon^p_{loosened} = 0.10$ (in shear).

### 3.3. The Simulation Procedure

We use FLAC$^{3D}$ software to simulate the longwall mining process and implement the proposed dynamic zoning method. The step-by-step excavation simulation is carried out, and in each step, 10 m of the coal seam is planned to be explored. In each step of the mining process, the simulation is conducted as Figure 7 shows, and temporary stability is expected in each step. After achieving the stability of the calculation model, the calculation results are recorded, including the displacement of all nodes and the state of all elements, and then simulation of the next step of the mining process begins.

### 3.4. Numerical Simulation Results

3.4.1. Progressive Caving Caused by Longwall Mining

In the process of longwall mining, coal-seam roof caving can be divided into two phases, i.e., the first caving and the periodic caving. When the length of the mined-out area

is short, the coal-seam roof will not collapse at once. When the length of the mined-out area reaches $L_0$, the first caving occurs due to the impact of gravity. With the advancement of mining, the coal-seam roof begins to bend down like cantilever beams. With a critical length of $Lp$, the cantilever beams break and fall. Periodic caving occurs with each advance of $Lp$ until mining is completed.

Figure 13 shows the roof caving of coal seams when the mining face extends to positions at 20, 30, 40, 50, 60, 70, 90, and 100 m. Figure 13 shows that when the mining reaches the 20 m position, the roof of the coal seam begins to bend (Stage 1). When mining to a position of 30 m (Phase 2), the first roof caving of the coal seam occurs ($L_0 = 30$ m). When further mining reaches the 40 m position (Phase 3), the coal-seam roof bends again, and when mining reaches the 50 m position (Phase 4), the coal-seam roof collapses for the second time ($Lp = 20$ m). Before the completion of longwall mining, with the advancement of $Lp$ ($Lp$ 10~30 m), repeated roof falls occur.

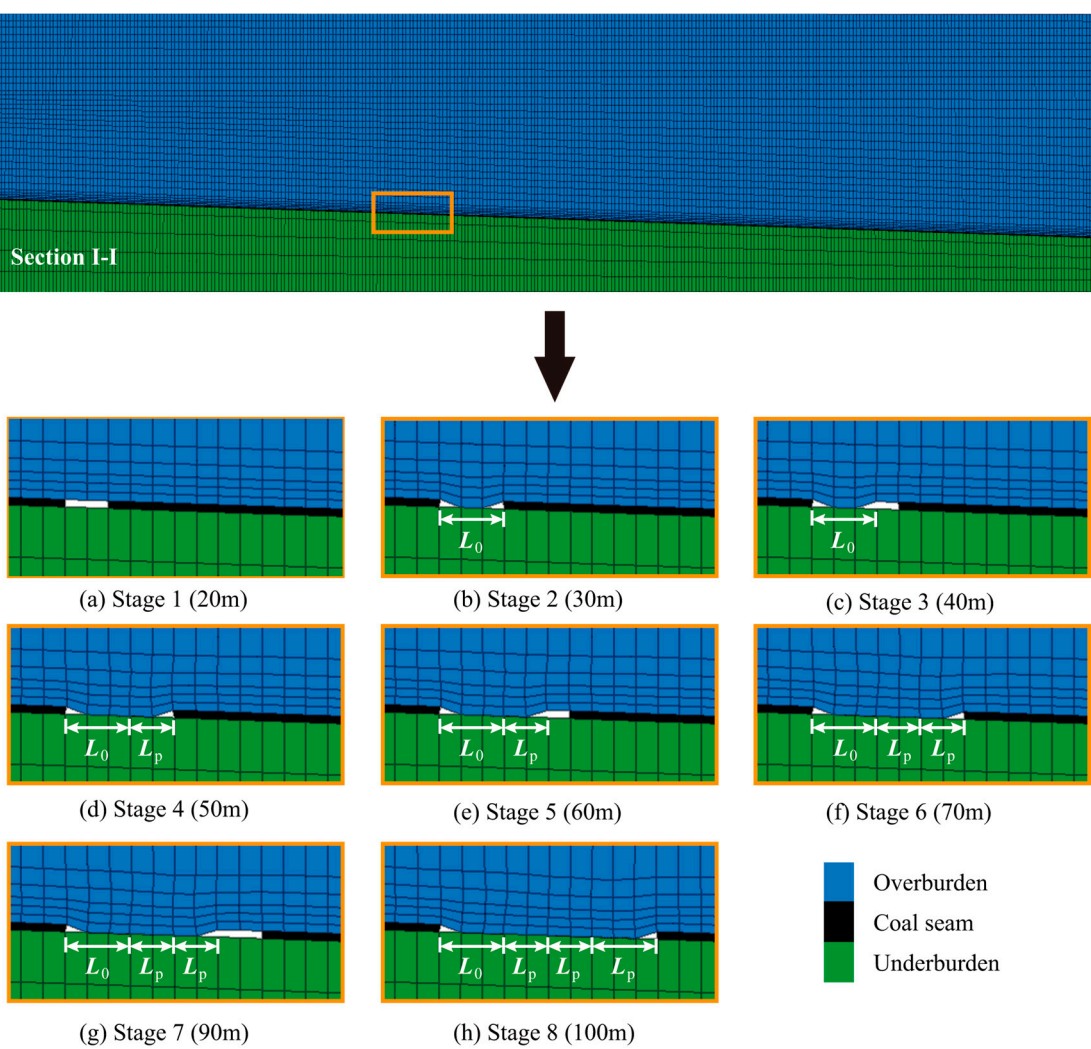

**Figure 13.** Simulated progressive caving of the overlying strata due to excavation.

The above process indicates that the roof of the coal seam will not collapse immediately after completing the corresponding mining steps. When the mining length is relatively short, the roof of the coal seam will bend downward due to the influence of gravity. When the advancing distance reaches a certain limit ($L_0$ or $Lp$), the roof of the coal seam begins to crack and collapse.

### 3.4.2. Dynamic Development Characteristics of the Four Zones in Longwall Mining

Figure 14 illustrates the dynamic development characteristics of the four zones when the excavation face is extended to the positions of 20 m, 30 m, 40 m, 50 m, 60 m, 70 m, 90 m, and 100 m. At first, the impact of the longwall mining on the overlying strata was quite small. According to the principal plastic strain $\varepsilon^p$, the overlying disturbed rock masses were in the post-peak softening stage. When mining to the position of 30 m, the scope of the overlying disturbed rock masses was suddenly expanded while the first roof caving occurred. From bottom to top, the overlying disturbed rock masses were in the stage of caved material deformation (Stage IV), residual strength (Stage III), and post-peak softening (Stage II). According to the zoning criteria, the overlying disturbed rock masses belong to the caved, fractured, and continuous deformation zones. The scope of the overlying disturbed rock masses was expanded periodically when further mining to the positions of 50 m, 70 m, and 100 m. Correspondingly, the four zones periodically expanded upwards and forwards.

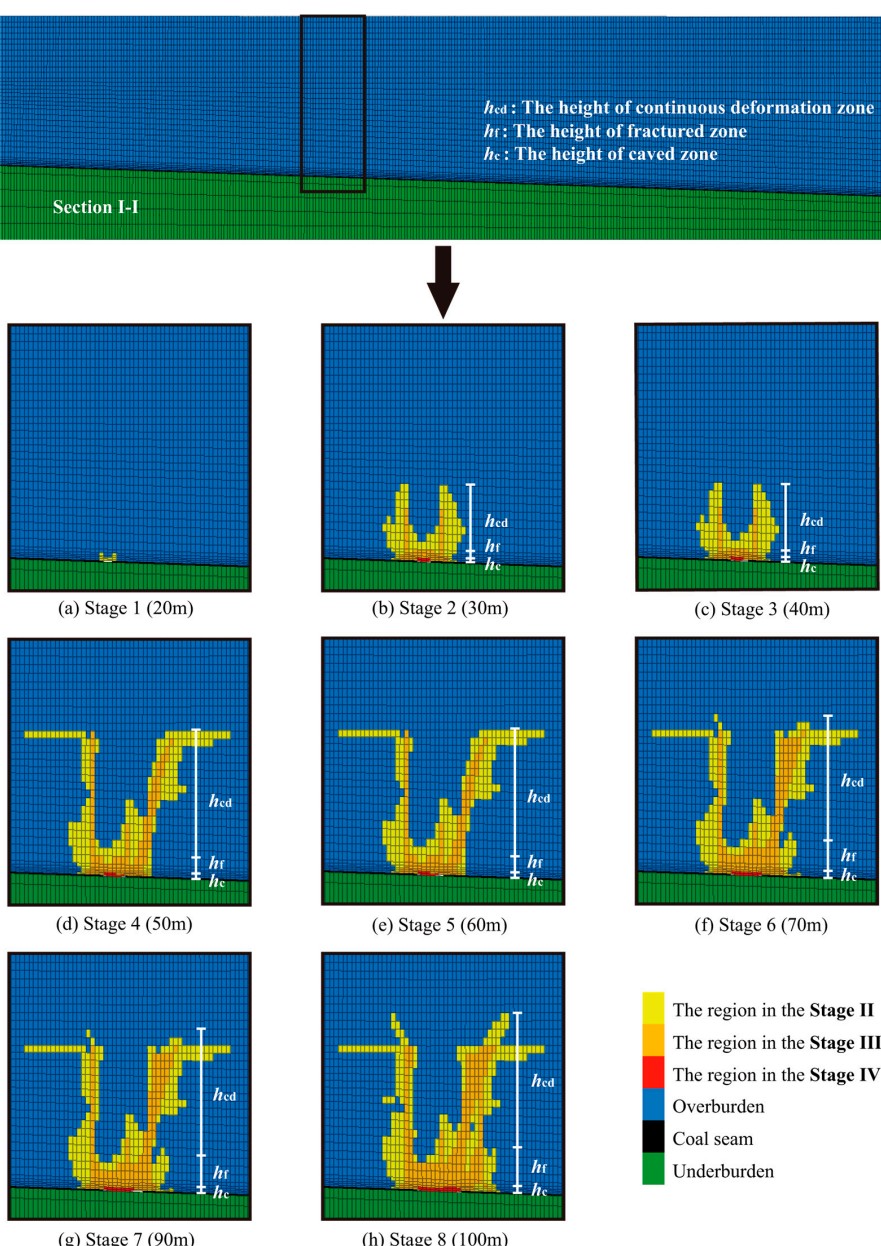

**Figure 14.** Dynamic evolution laws of the four zones due to excavation (20–100 m).

With the advance of longwall mining, the scope of the overlying disturbed rock masses is gradually expanded (see Figure 15). When mining to 200 m, the influence of the excavation developed to the ground surface, and the four zones were formed. As shown in Figure 16, when mining further, the heights of the four zones are nearly constant, and the four zones will only develop along the mining direction. After the completion of longwall mining, the heights of the four zones are approximately 12 m, 102.8 m, 380 m, and 50 m, respectively.

The above process indicates that the formation of the four zones is a dynamic process and of strong relevance to coal-seam roof caving. With the advancement of the excavation face, the coal-seam roof periodically caves, and the scope of the overlying disturbed rock masses expands periodically. The four zones will also periodically expand upwards and forwards. When the mined-out area reaches a certain extent, the heights of the four zones are nearly constant at maximum, and then the four zones will only develop along the mining direction.

In addition, as shown in Figures 14 and 15, the advance influence of longwall mining on the overlying strata and the surface in front of the excavation face can be observed in the mining process. The overlying strata and surface in front of the mined-out area are influenced by mining, and the closer to the excavation face, the more significant the influence of the longwall mining on overlying strata and surface. As shown in Figure 17, surface cracks are first formed around the excavation boundary, and when mining further, surface cracks regularly develop along both directions that are parallel to and perpendicular to the direction of mining.

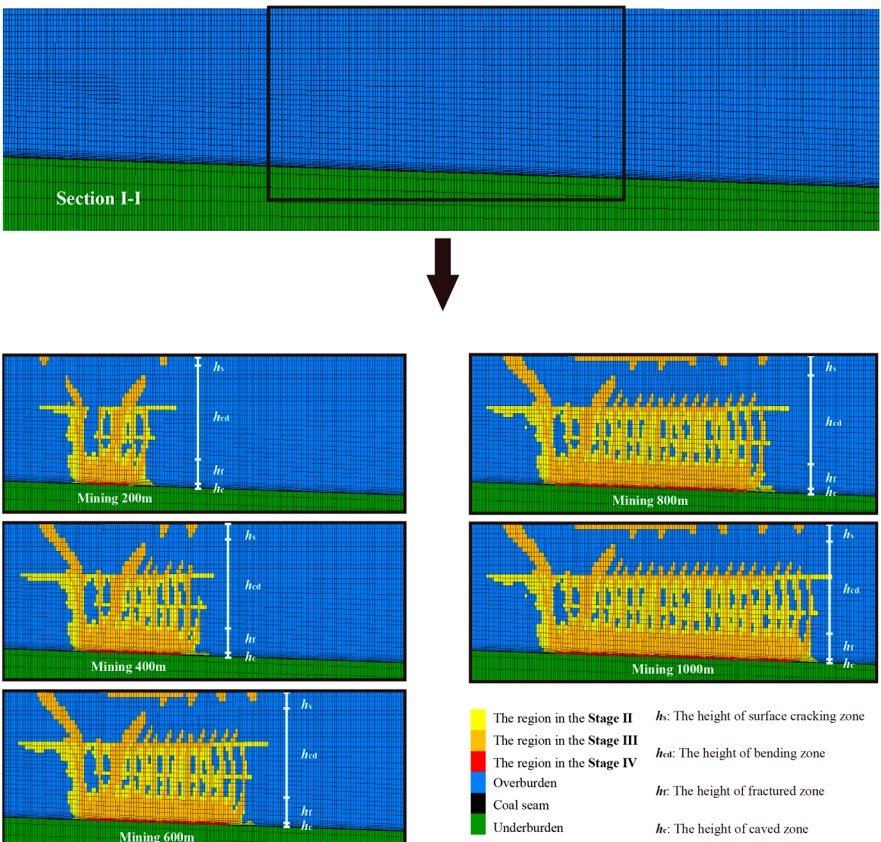

**Figure 15.** Evolution laws of the four zones due to excavation (200 m, 400 m, 600 m, 800 m, and 1000 m).

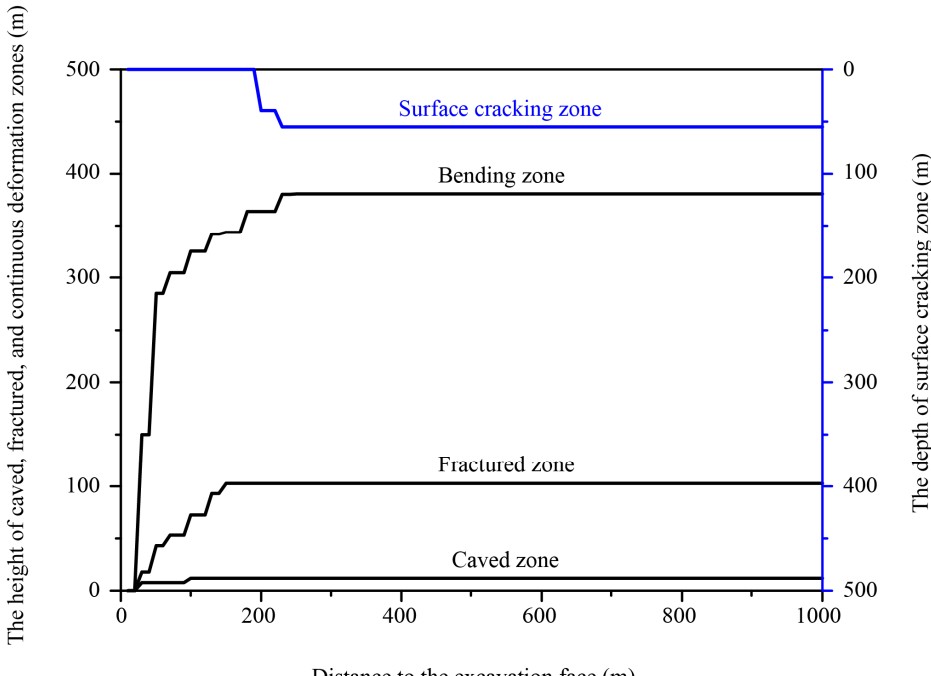

**Figure 16.** Relations between the advancing distance and the heights of the four zones.

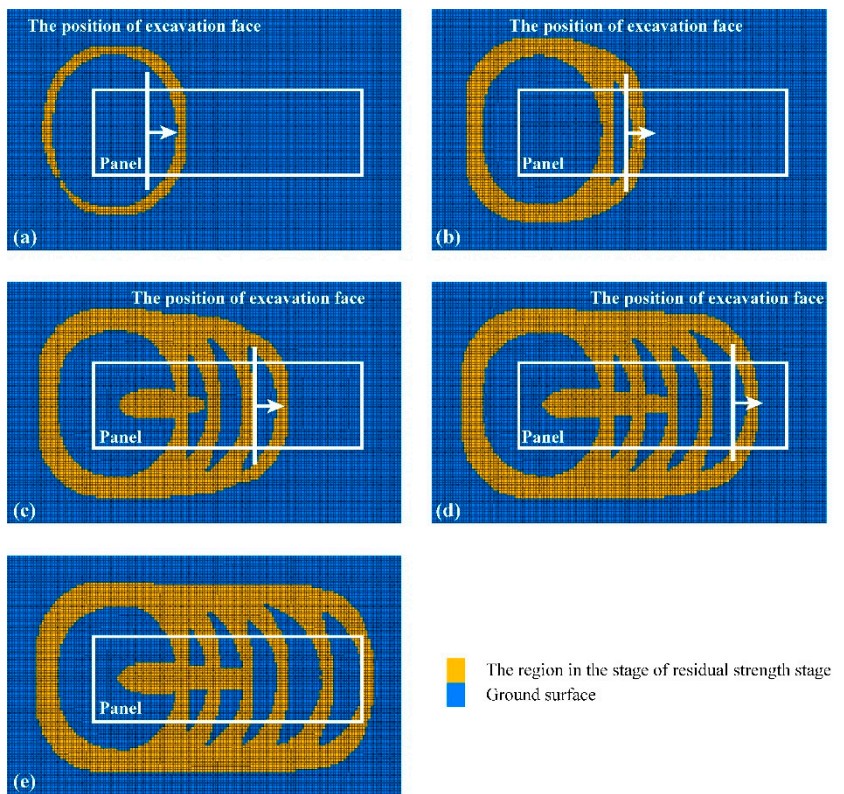

**Figure 17.** Top view of surface damage ((**a**) 200 m, (**b**) 400 m, (**c**) 600 m, (**d**) 800 m, (**e**) 1000 m).

### 3.4.3. Ground Surface Movement

Figure 18a,b show contour plots of horizontal surface displacement after completion of longwall mining. It can be observed that (1) the contour of the horizontal displacement is roughly parallel to the boundary of the mining area; (2) horizontal displacement above

the center of the goaf is relatively small; and (3) maximum horizontal displacement occurs in the surface area that is actually located above the boundary of the goaf.

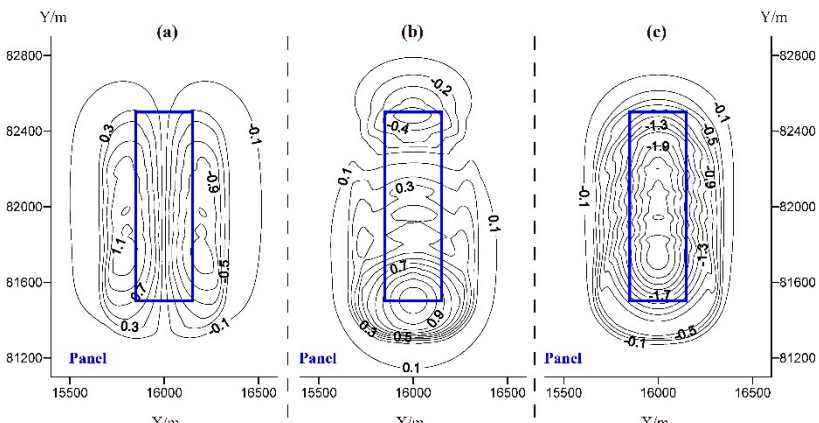

**Figure 18.** Contour maps of surface displacement: (**a**) X direction; (**b**) Y direction; (**c**) Z direction.

Figure 18c shows a contour map of surface subsidence (i.e., vertical displacement) after completion of longwall mining. It can be observed that (1) the contour of surface subsidence is roughly parallel to the boundary of the mining area; (2) maximum surface subsidence (2.86 m) occurs in the area above the center of the goaf; and (3) surface subsidence gradually decreases from the area above the center to the boundary of the goaf.

As shown in Figure 18, the final subsidence basin could be roughly divided into three regions: central region, compression region, and extension (tensile) region. In the central zone, the settlement is evenly distributed and can reach the maximum value, while in contrast, the horizontal displacement is very small. A compression zone is an area located between the boundary of a subsidence basin and the central zone, where subsidence is unevenly distributed, resulting in compression. The tensile zone is located near the boundary of a subsidence basin, where subsidence gradually decreases to zero and always produces tensile deformation due to horizontal surface displacement moving towards the center of the subsidence basin.

### 3.4.4. Risk Assessment

During the mining of Coal Seam # 3, the laying of the Quaternary aquifer may affect the mining safety. The water source of the Quaternary aquifer is atmospheric precipitation and surface water. The underlying stratum of the Quaternary aquifer is Neogene mudstone, which has a certain water resistance.

The average distance from the Upper Tertiary Aquifer to Coal Seam 3 is approximately 193.8 m. As shown in Figure 17, the total height of the depression and fault zone is about 114.8 m, which means that the fractures in the fault zone will not pass through the overlying aquifers, causing little damage to mining safety.

However, on the southern side of the continuous deformation zone, the crack height is large. This means that the fissure may completely pass through the Neogene aquifer and be connected to the Quaternary aquifer. Therefore, in this case, groundwater can seep into the goaf through cracks. In addition, air pressure along cracks can lead to air leakage and may harm mining safety.

According to height prediction results of the four areas, areas where (1) wide surface cracks and (2) high cracks may occur on the ground should be filled in a timely manner during observation (such as bulldozer rolling) to avoid air leakage and water gushing.

## 4. Discussion

The four zones of overlying disturbed rock strata are dynamically developed during the mining process, and the mechanical properties in different zones are correspondingly

changed. The dynamic zoning characteristics of disturbed rock masses should not be neglected. Otherwise, the numerical simulation will lead to deviations from the actual results. To predict the dynamic development characteristics and the height of the four zones more accurately, an FDM-based dynamic zoning method for the disturbed rock masses above the longwall mining panel is proposed. The proposed method is applied to the Taixi coal mine, and the dynamic development characteristics of the four zones of overlying disturbed rock masses are analyzed and predicted according to the dynamic zoning results.

The coal-seam roof above the mined-out area does not fall immediately with the advancement of the excavation face but caves periodically during the mining process. When the excavation face advances a short distance, the coal-seam roof can be considered a cantilever beam (cantilever plate), and bending deformation occurs due to gravity. When the excavation face advances to 30 m, the coal-seam roof caves for the first time. With the advance of $L_p$ ($L_p$ = 10~30 m), the coal-seam roof caves periodically until the end of mining. The periodic caving characteristics of the coal-seam roof conform to the caving phenomenon in actual mining [2] and other research work [3]. More specifically, Gao et al. [3] used UDEC to reveal the progressive caving of strata above a longwall panel. Their numerical results indicated that the immediate roof acts like beams and collapses periodically. The features of progressive caving fit reasonably well with the field observations in the Ruhr coalfield. Wang et al. [11] conducted a physical modeling experiment to simulate the overlying strata destruction induced by mining and revealed the different roof failure zones. Their results showed that the overburden failure evolution was accompanied by periodic collapse in the layer group and that the developing height of fractures discontinuously jumps.

The formation of the four zones is a dynamic process and strongly relevant to coal-seam roof caving. With the advancement of the excavation face, the coal-seam roof periodically caves, and the four zones will also periodically expand upwards and forwards. When the mined-out area reaches a certain extent, the heights of the four zones are nearly constant at maximum, and then the four zones will only develop along the mining direction. The development characteristics of the four zones are in accordance with results in the literature [6,14,27].

After finishing the entire mining process, the heights of the caved and fractured zones are 12 m and 102.8 m, respectively. By using the empirical formula [1] to calculate the maximum heights of the caved and fractured zones, the calculated maximum heights are 7~28 m and 105~175 m, which are consistent with the numerical modeling results.

After the completion of mining, the final subsidence basin can be roughly divided into three subzones, i.e., the central zone, the compressive zone, and the tensile zone. The maximum surface subsidence is approximately 2.86 m, and the subsidence coefficient is 0.82 (Equation (4)). As the Taixi coal mine has not yet been mined, the measured surface subsidence cannot be obtained. There are several coal mines close to the Taixi coal mine with similar geological conditions, including the Yanzhou coal mine (80 km away from the Taixi coal mine in the southeast), Zaozhuang coal mine (140 km away from the Taixi coal mine in the southeast), and Fengfeng coal mine (180 km away from the Taixi coal mine in the west). The subsidence coefficients of these neighboring coal mines are in the range of 0.78~0.88 (Table 2). Compared with the settlement coefficient obtained from several neighboring mines in Shandong Province (Table 2), the settlement coefficient calculated by Taixi Coal Mine is reasonable:

$$q = \frac{W_{cm}}{M \cdot \cos \alpha} \tag{4}$$

where $W_{cm}$ is the maximum value of subsidence, $M$ is the mining thickness, $q$ is the surface subsidence coefficient, and $\alpha$ is the dip angle of the coal seam.

**Table 2.** The measured subsidence coefficients of neighboring coal mines.

| Mining Area | Thickness (m) | Dip Angle (○) | Mining Method | $\sigma_{c\text{-}average}$ * (MPa) | Subsidence Coefficient |
|---|---|---|---|---|---|
| Fengfeng | 0.8<br>2.4 | 19<br>11 | Longwall | 47.7<br>57.9 | 0.78<br>0.84 |
| Zaozhuang | 1.5<br>1.9 | 24<br>4 | Longwall | -<br>- | 0.88<br>0.78 |
| Yanzhou | 7.8<br>8.2 | 4<br>4.3 | Fully mechanized caving | 23.1<br>13 | 0.81<br>0.84 |
| | 0.9<br>8.5 | 6.5<br>4 | Longwall | 24.5<br>22 | 0.80<br>0.83 |

* $\sigma_{c-average}$ is the average uniaxial compressive strength of the overlying strata.

The above comparison shows that the numerical simulation results obtained by this work, such as the periodic caving of the coal-seam roof and the dynamic development characteristics of the four zones, are consistent with the observed distributions and other research results. The heights of the caved and fractured zones are basically consistent with the empirical formula. The calculated subsidence coefficient is similar to the measured subsidence coefficients of neighboring coal mines. Thus, the dynamic zoning method is capable of analyzing and predicting the dynamic development characteristics of the four zones.

Compared with other zoning methods, the dynamic zoning method proposed in this paper specifically considers the dynamic zoning characteristics of disturbed rock masses in the longwall mining process to achieve more precise numerical results. More specifically, in the proposed method, the complete stress-strain curves of the disturbed rock mass are first simplified into four stages. Then, according to the simplified complete stress-strain curves, the zoning criteria are established by considering the deformation and failure characteristics of the disturbed rock masses, and the mechanical parameters of the disturbed rock masses are adaptively adjusted. Finally, the dynamic zoning of the disturbed rock masses in the mining process is effectively simulated on those bases, and the dynamic development characteristics of the four zones are obtained.

The influence of deformation and failure of the overlying disturbed rock masses on the mechanical parameters are considered, which improves the numerical modeling precision. However, it should be noted that in the proposed method, only the cohesion and internal friction angle of the disturbed rock masses are adjusted during the longwall mining process. Therefore, future work has been planned to adjust the elastic modulus and Poisson's ratio by considering the effect of deformation and failure of disturbed rock masses.

## 5. Conclusions

This paper proposes an FDM-based dynamic zoning method for disturbed rock masses above a longwall mining panel. This method is mainly composed of four stages: (1) establishing a simplified complete stress-strain curve; (2) determining the zoning criteria; (3) adaptively adjusting the mechanical parameters of the disturbed rock mass; and (4) numerically modeling the longwall mining based on the FDM.

To demonstrate the effectiveness, the proposed method has been applied to simulate underground mining in the Taixi coal mine. It has been concluded that (1) the coal-seam roof above the mined-out area does not fall immediately with the advancing of the excavation face but caves periodically; (2) the "four zones" of overlying strata are dynamically developed during the mining process and are strongly related to the roof-caving process; and (3) during the mining process, there is a high probability that the fissures in the fractured zone at the south side of the panel would connect with the fissures in the surface cracking zone, which needs careful attention and specific treatments (for example, by bulldozer rolling) in the actual longwall mining process.

The numerical simulation results obtained in this work, such as the periodic caving of coal-seam roof and the dynamic development characteristics of the four zones, are consistent with the observed distribution and other research results. This indicates that the proposed dynamic zoning method is capable of effectively analyzing and predicting the dynamic development characteristics of the four zones.

In the proposed method, only the cohesion and the internal friction angle of disturbed rock masses are adjusted during the longwall mining process. Future work will be focused on estimating and adjusting the elastic modulus and Poisson's ratio of disturbed rock masses.

**Author Contributions:** Conceptualization, S.J.; methodology, S.J.; validation, K.Z.; formal analysis, K.Z.; writing—original draft preparation, K.Z.; writing—review and editing, K.Z.; supervision, S.J.; project administration, S.J.; funding acquisition, S.J. All authors have read and agreed to the published version of the manuscript.

**Funding:** This research was funded by the National Natural Science Foundation of China, grant number 42230709, 41772326.

**Data Availability Statement:** Not applicable.

**Conflicts of Interest:** The authors declare no conflict of interest.

## Appendix A

**Algorithm A1.** Algorithm Code of Dynamic Rock Parameter Adjustment.

```
loop while p_z # null

    id=zone.id(p_z)
    w=zone.extra(p_z,11)

    if zone.model(p_z)# 'null' then
        if zone.group(p_z)#'50' then
            if zone.group(p_z)#'2' then
                if zone.group(p_z)#'1' then
                    if zone.group(p_z)#'0' then
                        ai1=zone.strain.inc.xx(p_z)
                        ai2=zone.strain.inc.yy(p_z)
                        ai3=zone.strain.inc.zz(p_z)
                        ai4=zone.strain.inc.xy(p_z)
                        ai5=zone.strain.inc.yz(p_z)
                        ai6=zone.strain.inc.xz(p_z)
                        dum2= zone.stress.prin.dir(p_z,as,ad)
                        l=ad(1)/math.sqrt(ad(1)*ad(1)+ad(2)*ad(2)+ad(3)*ad(3))
                        m=ad(2)/math.sqrt(ad(1)*ad(1)+ad(2)*ad(2)+ad(3)*ad(3))
                        n=ad(3)/math.sqrt(ad(1)*ad(1)+ad(2)*ad(2)+ad(3)*ad(3))
                        e1 = -(l*l*ai1+m*m*ai2+n*n*ai3+2*l*m*ai4+2*m*n*ai5+2*l*n*ai6)
                        if e1 > e1_max(id) then
                            e1_max(id) = e1
                        endif
                        if zone.state(p_z,1)=0 then
                            ee(id) = e1_max(id)
                        endif
```

**Algorithm A1.** *Cont.*

```
if zone.state(p_z,1)>0 then
    zone.extra(p_z,1) =-as(1)/1e6;;;;sigma1
    zone.extra(p_z,2) =-as(2)/1e6;;;;sigma2
    zone.extra(p_z,3) =-as(3)/1e6;;;;sigma3
    sigma_1=zone.extra(p_z,1)
    sigma_3=zone.extra(p_z,3);;;;sigma_1>sigma_3
    sig_1=as(1)/1e6
    sig_3=as(3)/1e6;;;;sig_1<sig_3
    sig_t=zone.prop(p_z,'tension')/1e6

nfri=(1+math.sin(zone.prop(p_z,'friction')*math.degrad))/(1-math.sin(zone.prop(p_z,'friction')*math.degrad))
    ap=math.sqrt(1+nfri*nfri)+nfri
    sig_p=sig_t*nfri-2*zone.prop(p_z,'cohesion')*math.sqrt(nfri)/1e6
    h=sig_3-sig_t+ap*(sig_1-sig_p)
    if zone.group(p_z)# 'fracture_zone_sr' then
        if zone.group(p_z)# 'fracture_zone_tr' then
            if h<0 then
                if ep(id)>0 then
                    if ep(id)<epr(id) then
                        zone.group(p_z)='fracture_zone_ss'
                        mm1=-mp(w,1)*mp(w,3)*ep(id)/epr(id)+mp(w,1)
                        nn1=-mp(w,2)*mp(w,4)*ep(id)/epr(id)+mp(w,2)
                        zone.prop(p_z,'cohesion')=mm1
                        zone.prop(p_z,'friction')=nn1
                    else
                        zone.group(p_z) = 'fracture_zone_sr'
                        msr=-mp(w,1)*mp(w,3)+mp(w,1)
                        nsr=-mp(w,2)*mp(w,4)+mp(w,2)
                        zone.prop(p_z,'cohesion')=msr
                        zone.prop(p_z,'friction')=nsr
                    endif
                endif
            endif

            if h>0 then
                zone.group(p_z)='fracture_zone_tr'
                mtr=-mp(w,1)*mp(w,3)+mp(w,1)
                ntr=-mp(w,2)*mp(w,4)+mp(w,2)
                zone.prop(p_z,'cohesion')=mtr
                zone.prop(p_z,'friction')=ntr
            endif
        endif
    endif

    if zone.group(p_z)='8' then
        if h<0 then
            zone.group(p_z) = 'failure-shear'
            zone.prop(p_z,'cohesion')=0.006e6
        endif
        if h>0 then
            zone.group(p_z) = 'failure-tension'
            zone.prop(p_z,'cohesion')=0.006e6
        endif
    endif
```

**Algorithm A1.** *Cont*.

```
                                    if w>0 then
                                        if zone.group(p_z) # 'caving_zone' then
                                            if zone.group(p_z) # 'failure_shear' then
                                                if zone.group(p_z) # 'failure_tension' then
                                                    if sigma_3<5 then
                                                        epr(id)=sigma_3*mp(w,5)+mp(w,9)
                                                    else
                                                        if sigma_3<10 then
                                                            epr(id)=sigma_3*mp(w,6)+mp(w,10)
                                                        else
                                                            if sigma_3<15 then
                                                                epr(id)=sigma_3*mp(w,7)+mp(w,11)
                                                            else
                                                                if sigma_3<20 then
                                                                    epr(id)=sigma_3*mp(w,8)+mp(w,12)
                                                                endif
                                                            endif
                                                        endif
                                                    endif
                                                endif
                                            endif
                                        endif

                                        if zone.group(p_z)='fracture_zone_sr' then
                                            if ep(id)>10 then
                                                zone.prop(p_z,'cohesion')=0
                                                zone.group(p_z)='fracture_zone_sr-0'
                                                if zone.pos.z(p_z)<-0.034012*(zone.pos.y(p_z)-81500)-486.424
                                                    zone.group(p_z)='caving_zone'
                                                endif
                                            endif
                                        endif

                                        if zone.group(p_z)='fracture_zone_tr' then
                                            if ep(id)>1 then
                                                zone.prop(p_z,'cohesion')=0
                                                zone.group(p_z)='fracture_zone_tr-0'
                                                if zone.pos.z(p_z)<-0.034012*(zone.pos.y(p_z)-81500)-486.424
                                                    zone.group(p_z)='caving_zone'
                                                endif
                                            endif
                                        endif

zone.prop(p_z,'tension')=0.1*2*zone.prop(p_z,'cohesion')*math.cos(zone.prop(p_z,'friction')*math.degrad)/(1-
math.sin(zone.prop(p_z,'friction')*math.degrad))   ;1/10*sigma_c
                                        zone.prop(p_z,'poisson')=(1-math.sin(zone.prop(p_z,'friction')*math.degrad))/2+0.01
                                    endif
                                endif
                            endif
                        endif
                    endif
                endif
p_z=zone.next(p_z)
endloop
end
```

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
