# Peer review of "An FDM-Based Dynamic Zoning Method for Disturbed Rock Masses above a Longwall Mining Panel"

_applsci, doi:10.3390/app13074336_

Round 1

Reviewer 1 Report

English of the paper should be improved

Author Response

Thank you for your constructive suggestions.

This manuscript has been edited for proper English language, grammar, punctuation, spelling, and overall style by one or more of the highly qualified native English speaking editors at AJE.

Reviewer 2 Report

This is an interesting paper presenting a new method for longwall excavation modeling. The methodology and the results are described quite clearly. One serious shortage is the description of how the caving process is modeled in the excavated zones.  By the interface contact? Somehow refilled with the weak material? Any other way? This is omitted in the paper and definitely should be commented.

Two pictures in figure 15 are missing. Minor editorial errors were highlighted in gray. 

Author Response

Responses to Reviewer # 2

Please note that the modifications made are marked in red in the revised manuscript.

Comment # 1:

This is an interesting paper presenting a new method for longwall excavation modeling. The methodology and the results are described quite clearly. One serious shortage is the description of how the caving process is modeled in the excavated zones. By the interface contact? Somehow refilled with the weak material? Any other way? This is omitted in the paper and definitely should be commented.

Response:

The following methods are used to simulate coal seam mining: first, before FLAC3D simulation mining, the interfaces are set at the contact surface between the coal seam and the roof, which can avoid the dragging effect on the contact surface when the coal seam is mined, and prevent the distortion of the grid displacement data near the interface, so as to realize the full collapse of the coal seam roof and the natural collapse to the floor, and ensure the numerical simulation effect; Then, according to the coal seam mining step, the coal seam units within the step are emptied (that is, set to null model) to simulate the current step coal seam excavation; Finally, iterate to the equilibrium state (the maximum unbalance ratio is less than 1e-5), save the calculation model, and then simulate the next mining step according to the above method until the coal seam excavation of the working face is completed.

Comment # 2:

Two pictures in figure 15 are missing.

Response:

Two pictures in figure 15 are added.

Author Response

Please note that the modifications made are marked in red in the revised manuscript.

Comment # 1:

Line 155: “elastic” instead of “elastic-plastic”

Response:

In line 155, "elastic-plastic" has been replaced by "elastic".

Comment # 2:

Line 214: It is assumed that the rock mass strength decreases to the residual strength at a plastic strain of 0.05~0.25, which is not consistent with the initial plastic strain 0.58 of N2m stratum in Figure 12.

Response:

I'm sorry, there is a statement error in Line 214 of the paper. The author wants to express that when the plastic strain reaches 0.05~0.25, the rock mass will fall in the Caved material deformation stage (Stage IV). The strain of 0.05~0.25 refers to the starting point of Stage IV, rather than that when the strain is between 0.05~0.25, the rock mass is in the Stage IV. Therefore, we change lines 212-215 as follow:

“It can be summarized that the cohesion and friction properties of rock mass are assumed to drop to a minimum residual value at a plastic shear strain larger than 0.05 (5% strain) and the tensile strength falls to a residual value of zero at a plastic tensile strain larger than 0.01 (1% strain).”

Comment # 3:

Line 277-281: Why make this assumption? The residual strength of rock mass under different confining pressures is different, and Line 161 also shows that the residual strength increases as the increasing confining pressure. It can also be seen from Figure 3 that the confining pressure has a certain influence on the rock strength.

Response:

The assumption here is redundant. We have deleted this assumption in the revised version.

Comment # 4:

There is no legend in Figure 15.

Response:

Legend has been added in Figure 15.

Comment # 5:

Pay attention to writing in native language.

Response:

The paper has been edited and revised by AJE.

Comment # 6:

How is the dynamic zoning in Figure 7 implemented? How to realize the adjustment of the rock parameter in the numerical code? The corresponding code should be given.

Response:

Figure 7 shows the algorithm flow of dynamic rock parameter adjustment. The program code of the algorithm is written by using the FISH language in FLAC3D software, and it is embedded in the numerical simulation calculation process to realize the dynamic adjustment of the mechanical parameters of the disturbed strata. See Appendix 1 for algorithm code.

Comment # 7:

Line 331: In the iteration process, even if the rock parameters are adjusted unreasonably, the computation can be also balanced (i.e., the maximum unbalanced force ratio is less than 1e-5). How to achieve the reasonable adjustment of rock parameters?

Response:

It should be noted that the iterative calculation here is only the calculation process of FLAC3D itself, and does not involve parameter adjustment. The parameters are adjusted according to the plastic strain after each mining step, and the iterative calculation is required for each mining step.

Comment # 8:

How are the rock mass parameters of each stratum determined? For example, the Shanxi formation (P1x) is composed of black mudstone and grayish white sandstone.

Response:

The physical and mechanical parameters of the rock masses in the study area, which were determined from physical and mechanical experiments on rock masses and evaluations of the rock mass quality.

Each formation often contains several rock strata with different lithology. For simplicity, we use the weighted average method to assign a set of parameters to each formation. We first determine the cohesion, internal friction angle, modulus of elasticity, Poisson's ratio, gravity and other parameters of the main rocks in each rock stratum according to the mechanical experiment, and then determine the parameters of each formation according to the thickness of the rock stratum and the above parameter values using weighted average method.

Comment # 9:

Figure 14(h): In the height of hcd, there are many rock masses corresponding to the Stage â…¢. Why is it divided into continuous deformation zone?

Response:

The term "continuous deformation zone" is not suitable for this manuscript. We changed the term "continuous deformation zone" to "bending zone". In the bending zone, the rock stratum mainly undergoes bending subsidence deformation, but plastic yielding may also occur locally. Therefore, the height of hcd is divided into bending zone.

Round 2

Reviewer 3 Report

(1) The figures in the text are not clear. Meanwhile, the format of the present manuscript remains to be further revised.

(2) Note that the text needs to be expressed in professional terms.

Author Response

Thank you for your comments. My answer is in Word
